# New horizons in live and dehydrated black soldier fly larvae usage: Behavioral and welfare implications in "Bianca di Saluzzo" cockerels

**Valentina Bongiorno[1], Edoardo Fiorilla[1]\*, Marta Gariglio[1], Valeria Zambotto[1], Eleonora Erika Cappone[1], Stefania Bergagna[2], Isabella Manenti[1], Elisabetta Macchi[1], Francesco Gai[3], Achille Schiavone[1]**

**1** Department of Veterinary Sciences, University of Turin, Turin, Italy, **2** Veterinary Medical Research Institute for Piemonte, Turin, Italy, **3** Institute of Sciences of Food Production, National Research Council, Turin, Italy

\* edoardo.fiorilla@unito.it

## Abstract

The literature on poultry welfare and behavior reports numerous promising effects derived from the administration of live or dehydrated black soldier fly larvae (BSFL) as environmental enrichment; however, their use in slow-growing male chickens has never been evaluated. To fill this gap, we divided a total of 144 Bianca di Saluzzo male chicks aged 39 days old into three experimental groups (six replicates, eight birds/replicate): control (C, no enrichment provided), dehydrated larvae (DL, 4.12% as fed), and live larvae (LL, 15.38% as fed), and reared until 147 days of age. Explorative and aggressive behavior patterns were analyzed, in concomitance with a tonic immobility test and the avoidance distance test, heterophile to lymphocyte -H/L- ratio), and excreta corticosterone metabolites (ECM) matrices. Overall, LL and DL supplementation were both effective at mitigating aggressive interactions among chickens (P < 0.05), while the frequency of indoor explorative behavior was lower in the LL group compared with C and DL (P < 0.01). By contrast, we found no differences in fear reduction during the avoidance distance test in the LL or DL groups compared with C (P < 0.05). We found a trend for the H/L ratio to be C < LL < DL (P = 0.051), and ECM concentrations were significantly lower in DL and LL groups compared with C (P < 0.001). In conclusion, the use of DL and LL as environmental enrichment have the potential to produce beneficial outcomes in slow-growing male chickens.

## 1. Introduction

Recent research confirms the promising use of black soldier fly larvae (BSFL) as environmental enrichment in poultry production. Slow-growing genotypes, and especially meat and dual-purpose local breeds, are often the first choice in developing countries [1] and for small-scale rearing productions. Their use also holds great promise for tackling the challenges related to climate change by virtue of their impressive environmental and nutritional resilience [2]. Moreover, slow-growing genotypes are generally characterized by greater reactivity

**Data availability statement:** All relevant data is available at the public repository Zenodo, https://zenodo.org/records/12581538#:~:text=10.5281/zenodo.12581537.

**Funding:** This paper is partially supported by the Poultrynsect project (H2020 ERA-NETs SUSFOOD2 and CORE Organic Cofund, under Joint SUSFOOD2/CORE Organic Call 2019, agreement No. 48) and the PRIMA programme under grant agreement No 2015, project SUSTAvianFEED. The PRIMA programme is supported by the European Union. The funders had no role in study design, data collection and analysis, decision to publish, or preparation of the manuscript.

**Competing interests:** The authors have declared that no competing interests exist.

compared with fast-growing commercial lines [2]. While this peculiarity may help avoid the locomotory problems associated with other productions, such as broiler chickens [3], a potential downside might be a higher frequency of aggressive events among birds. Since rearing systems for slow-growing breeds must manage these active bird temperaments over longer rearing cycles, aggressive behaviors related to hierarchy establishment and sexual maturity are likely to be more pronounced than in industrial rearing systems [4,5]. Given the environmental benefits and production potential of slow-growing breeds, there is a clear need to optimize rearing conditions that improve animal welfare and reduce aggression. Incorporating BSFL as environmental enrichment may offer an effective way to manage these challenges by potentially reducing aggressive behaviors and enhancing social stability within flocks. Therefore, this study hypothesizes that including BSFL in rearing systems for slow-growing poultry breeds will decrease aggressive behaviors and promote a more stable social environment, particularly during key developmental phases such as hierarchy formation and sexual maturity.

The genotype selected for this research was the Bianca di Saluzzo (BS), an Italian dual-purpose chicken breed, the extinction of which was averted just a few years ago (https://www.pollitaliani.it/en/razze/bianca-di-saluzzo/) thanks to the efforts of the University of Turin (Italy). The BS is now part of the Slow Food foundation presidium, an organization for food biodiversity promotion [6]. The breed is characterized by its white plumage, tending towards spotted pale-yellow or cream [7], a red comb and wattles, yellow to white earlobes and a yellow beak and feet. By virtue of its well-proportioned skeleton-muscles, structure and moderate dimensions, the breed is highly resilient to environmental stressors [8].

Live, moving insect larvae stimulate natural bird behaviors much more than inanimate objects and their provision has been shown to produce both direct (welfare and behavior) and derived (performance and health) benefits in poultry production [9,10]. However, the practical challenges related to the production and storage of live larvae are not insignificant. The use of DL may offer a suitable compromise: despite being inanimate, and thus stimulating bird interest to a lesser degree, they are nutrient-rich, enhance bird welfare, and their associated handling practices are much easier [10]. However, both the larva form (live *vs* dehydrated) and chicken genotype influence the potential benefits to be gained from larva provision. The first results on live BSFL provision in a poultry species were those published by Veldkamp and van Niekerk [11] on turkey poults. They showed an improvement in plumage coverage when 10% live BSFL were integrated into the diet formulation, and fewer incidents of aggressive interactions were recorded among five-week-old birds. Moreover, the study reported a trend for greater foraging activity in week-old larva-supplemented chicks compared with controls, although the opposite was observed at both three and five weeks of age. Interesting results were also obtained in the research conducted by Ipema and colleagues on broiler chickens [12,13]. The authors studied the effects of long-term live BSFL administration at both the 5 and 10% dietary inclusion level, and reported advantages in terms of bird activity levels, fear reduction, and leg health, as well as no negative outcomes on bird health parameters. The same authors also compared the effects of scattering dehydrated *vs* live BSFL in the bird pens at the 8% dietary inclusion level. They observed greater improvements in the active behaviors of birds provided the live BSFL compared with the dehydrated larvae [9]. Finally, Biasato *et al*. [14] obtained encouraging results in broilers receiving a 5% supplementation of either live BSFL or mealworms in terms of foraging behavior and activity levels, with no negative effects on feather condition, corticosterone metabolites, and leg health. Only one paper to date has investigated dietary supplementation (10%) with live BSFL on a slow-growing broiler [15]. The study explored behavioral and ethological parameters in both sexes, with an overall increase in locomotory and foraging activity, and reduced fear responses observed in supplemented females.

Various studies have been conducted on laying hens. Ruhnke *et al.* [16] analyzed ranging behavior in free-range laying hens and found no differences between birds receiving *ad libitum* dehydrated larvae *vs* controls. A second study revealed better feather condition scores in birds receiving a live BSFL supplement (10% inclusion level) compared with controls, and differences in ranging behavior, such as more time spent on the litter [17]. By contrast, Tahamtani *et al.* [18] did not find any significant differences in the novel object and open field tests in laying hens receiving *ad libitum* live BSFL supplementation (0, 10 or 20%). Finally, one experiment has been published on Muscovy ducks administered 5% BSFL or yellow mealworm (YM) larvae in addition to the diet [19]. Only the control groups presented a rise in aggressive behaviors over time, whereas the opposite was observed in YM groups. Both insect groups showed decreases in excreta corticosterone metabolites (ECM) and in the heterophile to lymphocyte (H/L) ratio.

This study investigated the hypothesis that LL provision would produce better results than their dehydrated equivalents in terms of animal-based indicators of welfare, namely greater bird exploratory behavior and a reduction in aggressive behaviors, and greater stress and fear diminishment. This was supposed on the basis that live larvae would attract the attention and interest of chickens more than immobile DL [9]. The effects of live and dehydrated larvae (LL and DL, respectively) supplementation on the welfare of slow-growing male chickens have not been previously investigated. Therefore, the objectives of this study were to examine the patterns of exploratory and aggressive behaviors in slow-growing male chickens when provided with live versus dehydrated larvae, and to assess any potential benefits for poultry welfare and behavior. Additionally, the research aimed to gather data on the behavioral differences associated with each supplementation type, with a focus on providing insights for medium and small-scale poultry farms, which typically raise slow-growing breeds. The goal was to identify the most effective environmental enrichment options for these farming contexts.

## 2. Materials and methods

### 2.1. Animals and experimental design

The experiment was carried out at the research facility of the University of Turin (north-west Italy, 44.88572, 7.68381), following its review and approval by the University of Turin's Bio-ethical Committee of the, Via Verdi 8, 10124, Turin (Italy) (Prot. n°814715).

Fertilized eggs were obtained from the BS breeders kept at the avian conservation facility at the University of Turin's pilot poultry plant. Following an initial incubation period, the eggs were sexed by a technical specialist and only the males selected. Organic farming procedures were applied through the whole trial [20], and the chicks were reared in an environmentally controlled brood until 38 days of age. Upon hatching, the chicks were subjected to a photoperiod of 23 h of light and 1 h of darkness on the first day, with subsequent 1 h/day decreases in the light period until 6 hours of darkness and 18 h of light were established. The detailed description of the experimental diet and BSFL composition and feeding program can be found in Fiorilla *et al.* [21]. Briefly, at the age of 39 days, a wing tag was applied to each chick and birds were selected based on the average live weight (LW, 316.8 ± 1.4 g on average). A total of 144 birds were allotted to 18 pens (2 × 3.2 m) covered with rice husks as litter (8 chickens/pen). From 63 days of age until the end of the trial, the birds had access to an outdoor space measuring 2.2 × 4.5 m and the housing conditions were characterized by natural ventilation and illumination. The experimental design entailed three dietary treatments: the control (C) groups received a basal diet in which soyabean meal was completely substituted with an alternative local source of protein (Maize meal 46.1%, Field bean 11.0%, Pea protein 10.8%, Barley 4.7%, Sunflower meal

9.5%, Maize gluten 11.6%, Soybean oil 1.6%; apparent metabolizable energy 2834.24 kcal/kg, crude protein 18.10%, ether extract 3.63%, crude fiber 4.80%); The DL groups received the C diet with dehydrated BSFL, and the LL groups received the C diet with live BSFL, both supplements calculated as 5% of the expected daily feed intake (based on dry matter (DM)). A total of 6 replicates/treatment were created, with 8 birds/pen (48 birds/dietary treatment). The experiment started at the start of the summer period (May 2022) and concluded at the beginning of the autumn (October 2022). The recorded environmental temperature ranged from a min. of 10°C to a max. of 37°C. Average temperatures were as follows: May, 20.5°C (min. 10°C; max. 31°C); June, 24°C (min. 15°C; max. 33°C); July, 26.5°C (min. 17°C; max. 36°C); August 27.5°C (min. 18°C; max. 37°C); September, 20°C (min. 10°C; max. 30°C); October, 17°C (min. 9°C; max. 25°C). The experimental period lasted for 108 days (from when the birds were aged 39–147 days), over which the LW and average daily feed intake (ADF) were recorded, and the average daily gain (ADG) and feed conversion ratio (FCR) calculated. At 147 days of age, two birds per replicate were euthanized (total 36 birds). The rearing conditions for the remaining birds remained unchanged until 174 days of age, when growth performance data were collected and second slaughter carried out. The chickens were slaughtered in accordance with European Council Regulation (EC) No 1099/2009, using electrical stunning to ensure compliance with animal welfare standards and humane practices throughout the process. Behavioral evaluations were conducted until 147 days of age only in order that the reduction in the number of birds/pen would not influence the data collected. The DL and LL supplements were provided at 11 a.m. daily (except Sundays) from when the birds were aged 39 to 147 days. Larvae were homogenously distributed on a feeder plate to ensure all birds equal access to them. The same plates were also placed into the C pens, making larvae provision the only difference between C and BSFL-fed birds [22]. As birds were in their developmental growth phase, the feed consumption of C groups was recorded weekly as the reference feed intake and used to estimate the amount of LL or DL to be administered to the experimental groups. Subcategories of sampling times were created to compare the various tests carried out over the course of the trial (T0 = 39–43 days of age; T1 = 91–92 days of age; T2 = 140–141 days of age). The only exception was that used for the video recordings, where T1, T2, and T3 correspond to 52, 94, and 136 days of age, respectively (Fig 1).

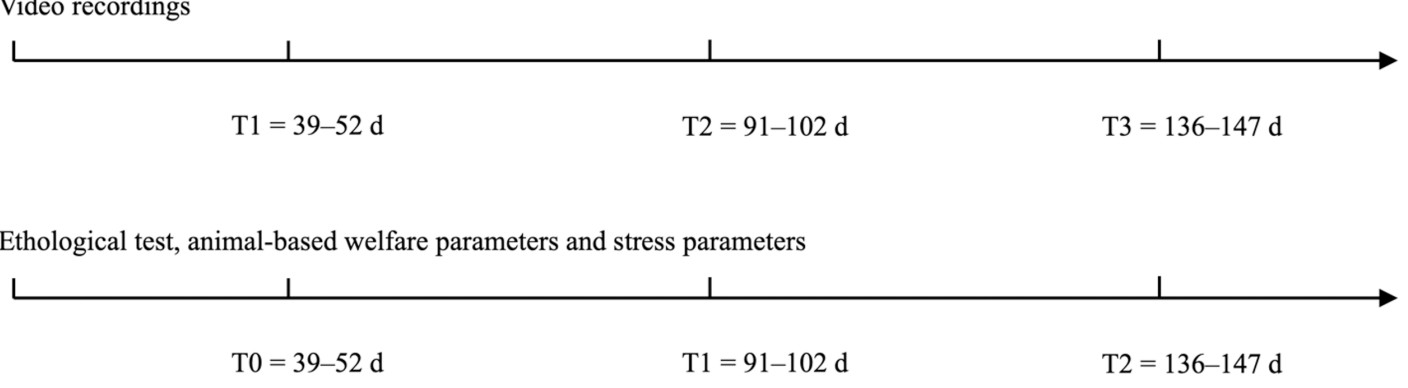

**Fig 1. Experiment timeline for video recordings, ethological test, animal-based welfare parameters and stress parameters conducted on a slow growing autochthonous chicken breed fed dehydrated and live black soldier fly larvae.** Abbreviations: T0, time 0; T1, time 1; T2, time 2; T3, time 3.

## 2.3. Behavioral observations

At 52 (T1), 94 (T2), and 136 (T3) days of age, the birds were videorecorded for behavioral assessment using cameras (SV3C, Tiger-Zhou UG, Germany) fixed to the pen ceilings. The time of BSFL administration was the focus of the behavioral observations, and three replicates/treatment were examined according to the ethogram reported in Table 1, a modified version of that used by Katajamaa et al. [23]. The software BORIS (Behavioral Observation Research Interactive Software v 7.9.7) [24] was adopted for the videorecording analysis. Explorative and agonistic behaviors were the two macro-categories considered. Explorative behavior comprised two categories: free exploration of the pen (ground pecking/scratching/object pecking) and larvae/plate (L/P)-related exploration. Agonistic behavior comprised the following categories: wing flap threat, raised hackle threat, chasing, aggressive pecking, and sparring/fighting behavior (Table 1). All behavioral data were collected from a 5-minute recording, which started upon the plate's insertion into the pen and considered the number of birds visible in the pen every 15 seconds. A behavioral event was counted once if the bird then disappeared from the camera's field of view for 10 or less than 10 seconds. Similarly, if a frequency behavior (e.g., shaking) not included in the categories analyzed, was preceded or followed by the same behavior of interest it was not counted twice. A total of one third of the video recordings were analyzed by a second operator trained by the most experienced observer; inter-rater reliability exceeded 0.90 for all parameters considered.

## 2.4. Ethological tests

The avoidance distance (AD) and tonic immobility (TI) tests were carried out by the same operator to assess fear levels in the birds. The TI test was carried out in a secluded area of the

**Table 1. Ethogram for evaluating the effects on behavior of feeding a slow growing autochthonous chicken breed dehydrated and live black soldier fly larva [22].**

| Macro-category | Category | | |
|---|---|---|---|
| Agonistic behaviors | Wing flap (Video A in S1) | Bird flaps wings < 0.5 m in front of other birds, regardless the body orientation of the other bird and the intensity of the flapping activity. | [19] |
| | Raised hackle (Video B in S1) | Body horizontal, head towards opponent, hackles of one (count as 1) or both (count as 2) raised. | Katajamaa et al., 2018 |
| | Chasing | Bird follows another bird, both running. | Katajamaa et al., 2018 |
| | Aggressive peck (Video C in S1) | Bird moves swiftly towards opponent miming to giving or gives aggressive peck. Head over opponent. | Katajamaa et al., 2018 |
| | Sparring/fighting (Video D in S1) | Bird involved in aggressive encounter. Peck/peck attempt, both birds active, running, jumping, or flying. | Katajamaa et al., 2018 |
| Indoor exploration | Plate/larvae exploration (Video E in S1) | Foraging behavior larvae/plate related (when larvae are not provided) (within 1 bird length from the plate, regardless bird orientation). Walking or standing, while approaching/head close to ground/plate, eyes focusing on ground/plate items, pecking at the ground/plate, moving litter backwards by means of the claws. | |
| | Free pen exploration (Ground pecking/ + scratching/ + object pecking) | Foraging activity NOT larvae/plate related (>1 bird length from the plate). Walking or standing, body axes with head towards/on the ground, moving litter backwards by means of the claws | Katajamaa et al., 2018) |
| Outdoor exploration | – | Explore the outdoor area (bird completely outside, thus not visible from the inside anymore) | |

rearing building to avoid any visual contact between the bird being tested and all other birds. If any events involving unusual noises occurred whilst performing a test, then that test was excluded from further analyses to make sure all data were collected under standard conditions in the absence of external disturbances. At the first sampling time (T0, 42 days of age), chickens from each group were randomly selected and labelled with a wing mark to make identifying them easier at the subsequent sampling times. The tests were repeated on the same birds at T1 and T2 (91 and 140 days of age, respectively).

In the TI test, in accordance with previous studies [25,26], each bird was laid on its back on a U-shaped cradle and the operator's hand pressed down, gently but firmly, upon the chicken's breast for 10 seconds to induce the immobility reflex; a stopwatch was used to record the time until the chicken had rightened himself; if the chicken did not rightened himself within 3 minutes, he was placed back on his feet by the operator [27]. A maximum of three attempts were made to induce the TI reflex before scoring the birds as 0. The attempt in which the TI reflex was obtained (first, second, third) was recorded and expressed as a percentage of the total number of attempts performed.

To evaluate the effect of providing birds with live or dehydrated BSFL on the fear response over time, we performed the avoidance distance (AD) test according to a version of the Welfare Quality® protocol [28]. The test was always carried out by the same operator, who maintained a squatting position at the entrance of the pen for 10 seconds. The number of chickens at three established distances from the operator (less than 1 m, 1–2 m, and over 2 m) was assessed at the end of the 10 seconds [22]. The test was always performed between 3.00 pm and 4.00 pm at the three sampling times: T0, T1 and T2.

## 2.5. Animal based welfare assessment

Animal-based welfare parameters were evaluated by means of a holistic approach. We evaluated plumage condition, the presence of bumble foot, and damage to the skin and comb using a modified version of the protocol devised by Tauson *et al.* [29]—specifically conceived to determine the welfare status of white laying genotypes. The scoring range adopted was 1 to 4 for all assessment, with the exception of footpad dermatitis which considered a range of 0 to 4, as proposed by Rushen *et al.* [30], based on the Welfare Quality® protocol [2009]. Higher scores implied higher welfare conditions as a reflection of lower injury severity. Overall, values < 2 indicated extensive feather pecking damage to the plumage, severe skin conditions and inflammation of the foot. The combined score for all parameters provided a complete and integrated evaluation of bird welfare (poor welfare conditions < 12; good welfare conditions > 18/20), and allowed us to quantify the extent of the damage and the body areas affected [27]. The observer, who performed all assessments, was trained using the photographic documentation provided in the respective protocols. All birds were assessed for all parameters at 39 (T0), 102 (T2), and 147 (T3) days of age. Feather damage encompassed the evaluation of six body areas, namely: neck, breast, wings, tail, back and cloaca/vent. The explanatory illustrations provided in the work by Tauson *et al.* [29] were used for training. Based on the documentation provided, the following technical table (Table 2) was composed:

As for plumage condition, the lesions derived from feather pecking were scored on the basis of the detailed pictures provided by Tauson *et al.* [29] (modified) (Table 2). Both bumble foot and footpad dermatitis were scored using the scoring systems proposed by Tauson *et al.* [29], Rushen *et al.* [30] and Welfare Quality® [28] (Table 2).

## 2.6. Excreta corticosterone metabolite analysis

Excreta samples were collected from the same two birds/pen (selected at random for the TI evaluation) at each sampling time (42 -T0-, 91 -T1- and 140 -T2- days of age). The birds

**Table 2. Scoring system used to determine feather pecking, wounds to skin and comb, bumble foot and footpad dermatitis.**

| Feather pecking | | |
|---|---|---|
| Score | Damaged/broken feathers, level (%) | |
| 1 | 100 | Extremely severe feather pecking |
| 2 | 50 | Severe feather pecking |
| 3 | 25 | Moderate feather pecking |
| 4 | 0 | Absence of feather pecking |
| Wounds | | |
| Score | Wound severity (cm) | |
| 1 | ≥ 2 | Very severe wound; presence of scab(s) and blood |
| 2 | 1–2 | Severe wound; visible scab/scabs; possible presence of blood |
| 3 | < 1 | Moderate wound; mostly irritated skin; possible scab(s) and presence of blood |
| 4 | 0 | No wounds |
| Bumble foot | | |
| Score | Damage level (% affected area) | |
| 1 | 100 | Extremely severe swelling |
| 2 | 50 | Severe swelling; scabs present |
| 3 | 25 | Light swelling; scabs present |
| 4 | 0 | No swelling or scabs |
| Footpad dermatitis | | |
| Score | Damage level (% affected area) | |
| 4 | 100 | Severe, deep damage; hyperkeratosis |
| 3 | 50 | Severe, deep damage; hyperkeratosis |
| 2 | 25 | Moderate, superficial damage; hyperkeratosis |
| 1 | 25 | Minor, superficial damage; hyperkeratosis |

were kept in individual wire-mesh cages (100 × 50 cm) until a total of 2 g of excreta had been produced by each bird. Excreta was collected in a plastic tray placed beneath the cage and each sample swiftly placed in Eppendorf tubes and stored at −20°C for the corticosterone analysis. ECM extraction followed the methods defined by Costa *et al.* [31] and Palme *et al.* [32]. In brief, 3 mL of 80% methanol (Sigma Aldrich, St. Louis, MO, USA) plus 0.25 g of lyophilized excreta were mixed in an extraction tube and maintained at −20°C for 2 h. After solid phase sedimentation, the supernatant was subjected to an evaporation process in another vial beneath a hood for 14 h. A multi species enzyme immunoassay kit (K014-Arbor Assay®, Ann Arbor, MI, United States) was used to determine the ECM. Inter- and intra-assay coefficients of variation < 10% were successfully obtained. The sensitivity of the assay was 11.2 ng/g of excreta. Repeated dilutions were performed prior to sample analyses (1:4, 1:8, 1:16, and 1:32), and all the regression slopes were parallel to the standard curve ($R^2 = 0.989$). The mean recovery rate of corticosterone added to dried excreta was 96.5%, whereas the kit's values of cross-reactivity, as reported by the manufacturer, were: 100% for corticosterone, 12.3% for deoxycorticosterone, 0.62% for aldosterone, 0.38% for cortisol, and 0.24% for progesterone. The samples were processed in duplicate, and the concentration expressed as ng/g excreta dry matter.

## 2.7. Heterophile to lymphocyte ratio

At 148 days of age, 2 birds/pen (36 birds in total) were selected on the basis of the average weight and sacrificed; blood samples were collected during bleeding. 2.5 mL of blood were stored in EDTA tubes. Smears were obtained by placing a drop of blood on each glass slide, then stained using May-Grünwald and Giemsa stains [33]. A 1:200 Natt-Herrick solution was used to treat the samples [34], and erythrocyte and leukocyte counts were carried out using an improved Neubauer hemocytometer [35]. A maximum of 100 granular (heterophils, eosinophils, and basophils) and non-granular (lymphocytes and monocytes) leukocytes were counted on the glass slide, and the H/L ratio consequently determined.

## 2.8. Statistical analysis

The software IBM SPSS Statistics [36] was used to perform the statistical analysis. Outlier analyses were carried out for the corticosterone values, but no behavioral data were excluded from the databases when biologically obtainable, and no anomalies were observed during the tests. The normality of data and residuals and the homogeneity of variances were assessed by means of Shapiro-Wilk's and Levene's tests, respectively. Three experimental groups were evaluated according to the enrichment provided: C, DL, and LL groups. Regarding the behavioral observations, a total of three replicates were observed for each enrichment group, while six replicates were considered for the AD and TI tests and for the animal-based welfare parameters. For all the above-mentioned parameters, the pen was considered the experimental unit (n = 3 for video recordings, n = 6 for all other tests). By contrast, the H/L ratio and ECM assessments (n = 12), and the TI duration-ECM correlations (n = 18) were designed considering the individual bird as the experimental unit. Furthermore, the animal-based welfare parameters, the AD test results, and the video recording data were analyzed using a Poisson loglinear distribution within a general linear mixed model (GLMM). Similarly, A GLMM was fitted to analyze the TI duration, TI induction frequency, and ECM, but following a gamma probability distribution and log-link function. The animal-based welfare parameters were analyzed considering a GLMM with a negative binomial response probability distribution and a nonlinear link function -log.

Finally, one-way ANOVA was used to analyze the H/L ratio; the data were found to be homogeneous after the Brown-Forsythe correction and the Duncan test was used as post-hoc test. The fixed factors, enrichment (E) and time (T), considered three (C, LL, DL-T0, T1, T2) levels, and the E × T interactions were assessed by means of pairwise comparisons, indicated by the replicate the measurement repetition of the same pen over time.

The mathematical formula adopted can be summarized as:

$$y_{ij} = \mu + \alpha_i + \beta_j + (\alpha\beta)_{ij} + \varepsilon_{ij}$$

$y_{ij}$ = dependent variable considering the fixed factors i (enrichment) and j (age, if present)
$\mu$ = overall mean.
$\alpha_i$ = Fixed effect of enrichment factor
$\beta_j$ = fixed effect of age factor (if present)
$(\alpha\beta)_{ij}$ = interaction between the fixed factors enrichment and age
$\varepsilon_{ij}$ = residual error unexplained in the model.
As neither the corticosterone nor TI data were normally and homogeneously distributed, correlations were determined using a Spearman correlation test. Data are shown as least square means and the standard error of mean (SEM), and P values ≤ 0.05 were considered as significant. Average observations < 0.5 times per sampling time were excluded from the statistical analysis.

## 3. Results

The growth performances of the BS males are reported in Table 3. The LW and ADG of the birds in the DL groups tended to be higher than in both LL and C, and the LW and ADG of birds in LL were higher than in C (P = 0.059 and P = 0.072, respectively) (Table 3).

### 3.1. Behavioral observations

**3.1.1. Indoor exploration behavior.** Overall, indoor exploratory behaviors were affected by time, enrichment, and their interaction. Specifically, no differences were observed between LL-DL and C groups, despite a higher frequency of indoor exploratory behavior in the DL groups compared with LL groups (P < 0.001) (Table 4). However, the frequency of indoor exploratory behavior was higher at T1 than at the other sampling times, whereas no differences were detected between T2 and T3 in either the DL or C groups (DL: 66.67 *vs* 25.00 and 33.33, P < 0.001 and P < 0.01, respectively; C: 53.67 *vs* 26.33 and 26.67, P < 0.001) (Fig 2).

In the LL groups, the frequency of free pen exploration behavior was only higher at T1 compared with T3 (36.67 *vs* 26.67; P < 0.05) (Fig 2). Regarding the comparisons between enrichment types at each sampling time, significant results were only obtained at T1, with the frequency of indoor exploration behavior being lower in C compared with DL groups (53.67 *vs* 66.67; P < 0.01) but higher compared with LL (53.67 *vs* 36.67; P < 0.01) (Fig 3).

**Table 3. The effects on growth performance of feeding a slow growing autochthonous chicken breed dehydrated and live black soldier fly larva (n = 6).**

|  | Diet | | | Age | | SEM | p-value | | |
|---|---|---|---|---|---|---|---|---|---|
|  | C | DL | LL | 147 d | 174 d |  | Diet | Age | D × A |
| LW (g) | 2335[b] | 2440[a] | 2412[a] | 2251 | 2541 | 20.605 | 0.029 | <0.001 | 0.848 |
| ADG (g/d) | 16.6[b] | 17.5[a] | 17.2[a] | 17.9 | 16.2 | 0.169 | 0.044 | 0.031 | 0.991 |
| ADFI (g/d) | 56.8b | 58.1a* | 58.6a* | 57.9 | 57.7 | 0.413 | 0.023 | 0.690 | 0.998 |
| FCR (g/d) | 3.51 | 3.37 | 3.48 | 3.24 | 3.67 | 0.046 | 0.114 | 0.019 | 0.544 |

Abbreviations: LW, live weight; ADG, average daily gain; ADFI, average daily feed intake (on a DM basis); FCR, feed conversion ratio (on a dry matter basis, including larva intake); C, control; DL, dehydrated larvae; LL, live larvae; SEM, standard error of the mean.

**Table 4. Effects on behavior of feeding a slow growing autochthonous chicken breed dehydrated and live black soldier fly larva, including enrichment treatment (E), time (T), and their interactions (E × T) (n = 3).**

| Item | Enrichment (E) | | | Time (T) | | | SEM | | P-value | | |
|---|---|---|---|---|---|---|---|---|---|---|---|
|  | C | DL | LL | T1 | T2 | T3 | E | T | E | T | E × T |
| Indoor exploration | 33.5[ab] | 38.2[a] | 31.1[b] | 50.8[a] | 27.5[b] | 28.5[b] | 2.62 | 2.10 | <0.001 | <0.001 | 0.022 |
| Plate/larvae exploration | 16.2[c] | 26.6[a] | 21.0[b] | 30.4[a] | 15.0[b] | 19.8[b] | 1.68 | 1.65 | 0.001 | <0.001 | 0.433 |
| Free exploration of pen | 16.6[a] | 10.9[b] | 10.1[b] | 20.2[a] | 11.4[b] | 7.93[b] | 1.77 | 1.37 | 0.001 | <0.001 | 0.003 |
| Outdoor exploration | Not detectable | | | | | | | | | | |
| Agonistic behaviors | 10.6[a] | 3.73[b] | 1.30[b] | 5.51[a] | 2.15[b] | 4.36[a] | 1.26 | 0.98 | <0.001 | <0.001 | <0.001 |
| Aggressive pecking | 2.55[a] | 1.87[ab] | 0.61[b] | 2.30 | 0.69 | 1.81 | 0.51 | 0.53 | <0.001 | 0.136 | <0.001 |
| Wing flapping | 2.42[a] | 1.18[b] | 0.00[c] | 0.00[b] | 1.01[a] | 1.48[a] | 0.25 | 0.39 | <0.001 | 0.004 | 0.034 |
| Raised hackle | Not detectable | | | | | | | | | | |
| Chasing | Not detectable | | | | | | | | | | |
| Sparring/fighting | Not detectable | | | | | | | | | | |

Abbreviations: C, control groups; DL, groups receiving supplement of dehydrated black soldier fly larvae; LL, groups receiving supplement of live black soldier fly larvae; T1, time 1; T2, time 2; T3, time 3. Superscript letters a, b, and c indicate significant differences at P ≤ 0.05.

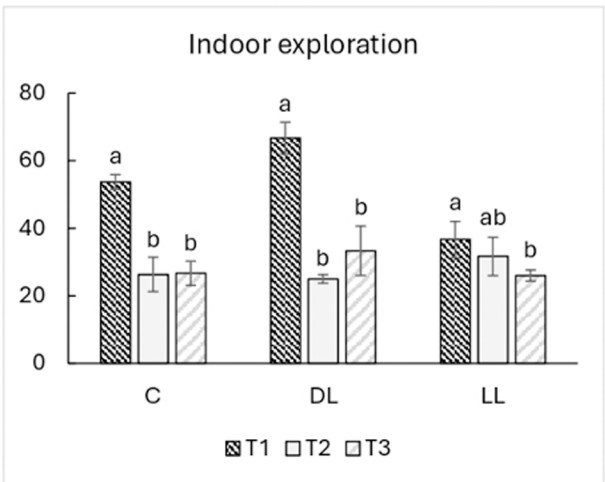
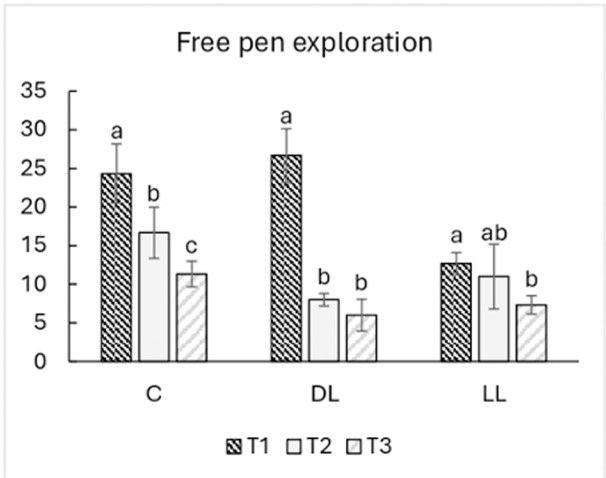
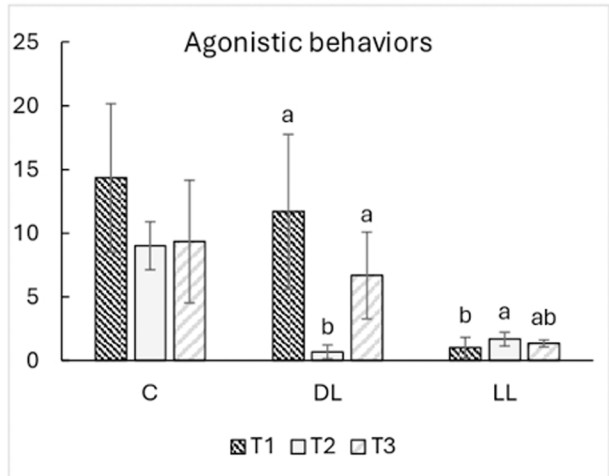
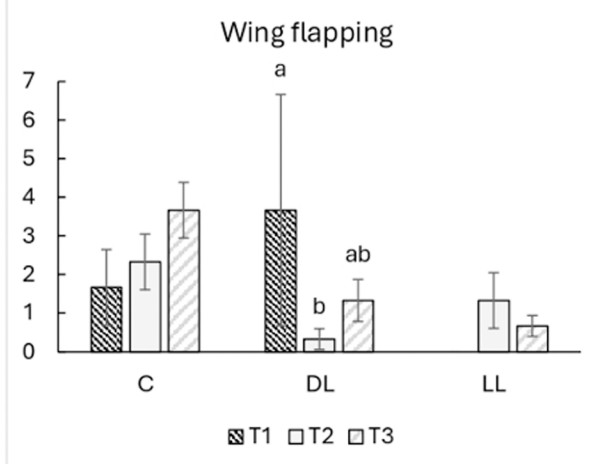
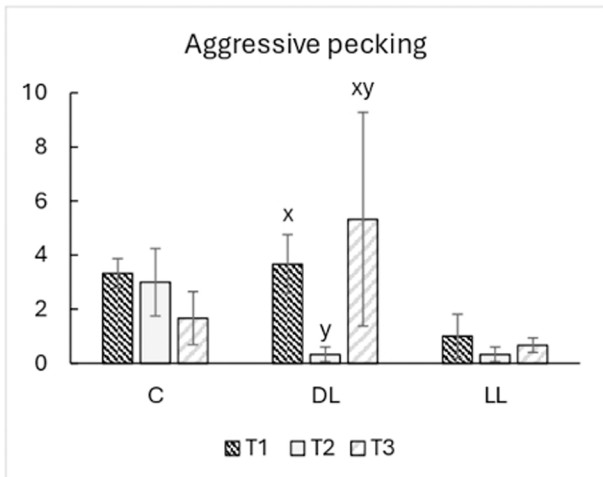

**Fig 2. The effects on behaviors of slow growing autochthonous chicken breed fed dehydrated and live black soldier fly larvae, including the interaction enrichment × time (E×T), and the differences between times within the same enrichment group (n = 3).** Abbreviations: T1, time 1; T2, time 2; T3, time 3; C, control groups; DL, groups supplemented with dehydrated black soldier fly larvae; LL, groups supplemented with live black soldier fly larvae. The letters a, b, and c are reported as unique graph label. The superscript letters a, b and c indicate significant differences ($P < 0.05$).

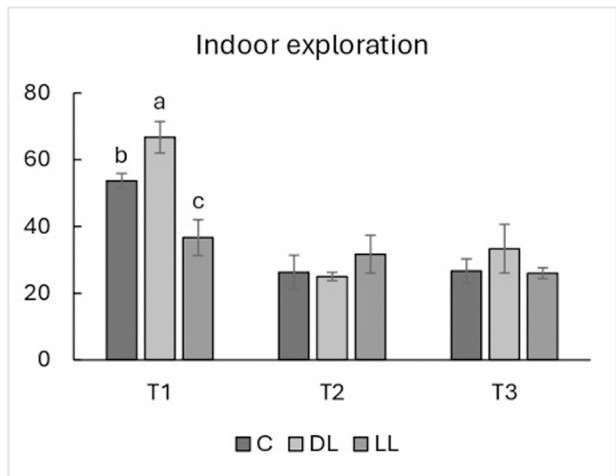
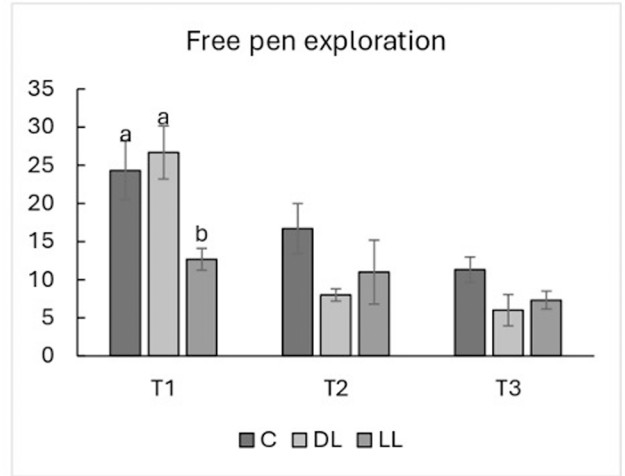
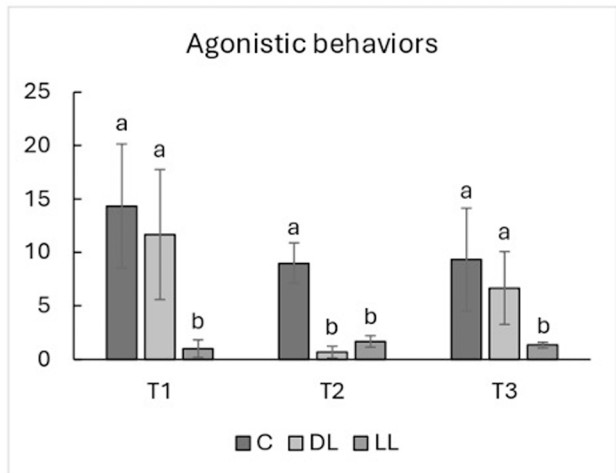
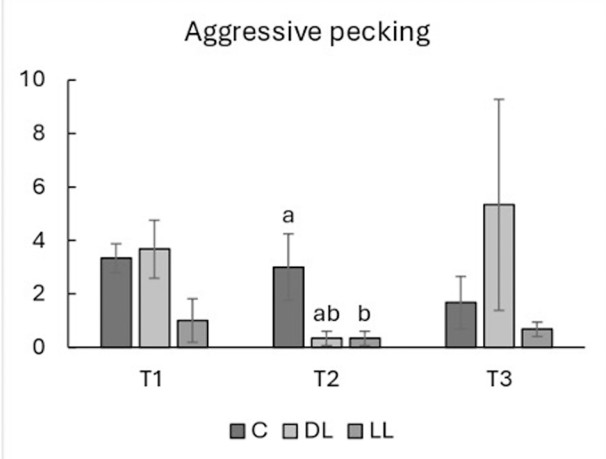
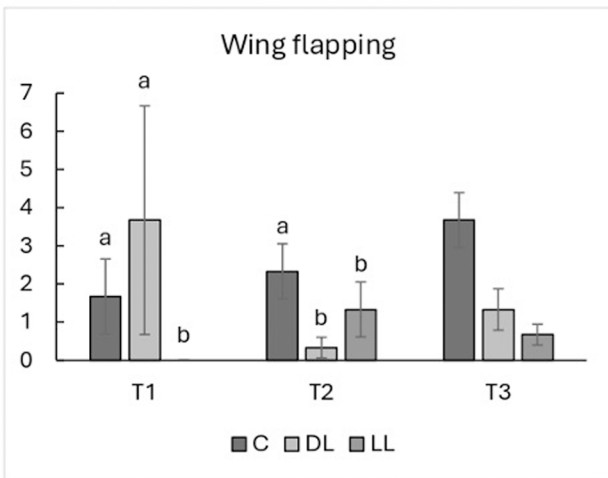

**Fig 3. The effects on behavior of a slow growing autochthonous chicken breed fed dehydrated and live black soldier fly larva, including the interaction enrichment × time (E×T), considering the differences between enrichment groups at sampling time (n=3).** Abbreviations: T1, time 1; T2, time 2; T3, time 3; C, control groups; DL, groups supplemented with dehydrated black soldier fly larvae; LL, groups supplemented with live black soldier fly larvae. The letters a, b, and c are reported as unique graph label. The superscript letters a, b and c indicate significant differences (P < 0.05).

Finally, the frequency of indoor exploratory behavior in the DL groups was almost double that exhibited by LL groups (66.67 *vs* 36.67; P < 0.01) (Fig 3). The different categories of indoor exploratory behavior, analyzed according to enrichment, time, and their interaction, accounted for the significant differences in both free pen exploration and L/P exploration, except for E × T interaction in L/P groups, where no statistical difference was found. Overall, the frequency of free pen exploration decreased over time in the C groups (24.33, 16.67, and 11.33; T1-T2, P < 0.001; T1-T3, P = 0.001; T2-T3, P < 0.05), whereas in DL groups a decrease in the frequency of this behavior was only detected between T1-T2 and T1-T3 (26.67 *vs* 8.00 and 6.00; P < 0.001) (Fig 2). By contrast, the frequency of free pen exploration in LL birds was higher at T1 than at T3 (12.67 *vs* 7.33; P < 0.001) (Fig 2). At T1, both C and DL groups showed a higher frequency of free pen exploration than LL (T1 24.33 and 26.67 *vs* 12.67; P = 0.001), whereas at T2 the values recorded for C groups were only higher than LL (16.67 *vs* 8.00; P < 0.05), and no differences were found between C-LL and DL-LL groups. Finally, at T3 the C groups presented more frequent free pen exploration compared with both DL and LL (11.33 *vs* 6.00 and 7.33; P < 0.001) (Fig 3). Regarding L/P exploration, the frequency of this behavior was higher at T1 than at both T2 and T3 (P < 0.001), while no differences were recorded between T2 and T3 (Table 4). Moreover, the frequency of the behavior was greater in both DL and LL compared with C groups (P < 0.01 and P < 0.05), but lower in LL than in DL groups (P < 0.001) (Table 4).

**3.1.2. Outdoor exploration.** The frequency of outside exploration behavior was insufficient to fit the statistical model and run the analyses.

**3.1.3. Agonistic behavior.** Overall, the macro-category encompassing agonistic behaviors was affected by the enrichment type, sampling time, and their interaction. More specifically, a higher frequency of agonistic behavior was observed in C than in DL (P < 0.01) and LL groups (P < 0.05), despite no differences being recorded between DL and LL (Table 4). An inconsistent pattern of agonistic behavior was observed considering the three sampling times, with higher frequencies noted at T1 *vs* T2 (P < 0.05), but lower at T2 *vs* T3 (P < 0.01); no significant differences were recorded between T1 and T3 (Table 4). Considering the E × T interaction between the sampling times, the DL groups displayed a higher frequency of agonistic behavior at T1 *vs* T2 (11.67 *vs* 0.67; P < 0.05), whereas it was lower at T2 *vs* T3 (0.67 *vs* 6.67; P < 0.05). Contrastingly, the opposite was recorded within the LL groups, which showed a less frequent agonistic behavior at T1 *vs* T2 (1.00 *vs* 1.33; P < 0.05) (Fig 2). Within each sampling time, the frequency of agonistic behavior was higher at T1 in C compared with LL (14.33 *vs* 1.00; P < 0.05). Similarly, DL groups tended to display a higher frequency of agonistic behavior than LL at T1 (11.67 *vs* 1.00; P = 0.096), whereas no differences were recorded between C and DL groups. At T2, both DL and LL groups showed a lower frequency of agonistic behavior than C (0.67 and 1.67 *vs* 9.00; P < 0.001). Finally, at T3 the frequency of agonistic behavior in LL groups tended to be lower than in C (9.33 *vs* 1.33; P = 0.078), while no differences were reported for the other parameters (Fig 3). Considering the different categories, only the data obtained for aggressive pecking and wing flapping were suitable for statistical analysis. Enrichment type significantly affected the frequency of aggressive pecking, being less frequent in LL birds than in C (P < 0.001), despite no significant differences being detected between DL and LL or DL and C (Table 4). A reduction in aggressive pecking was observed in the DL groups at T2 compared with T1 (3.67 *vs* 0.27; P = 0.001). Furthermore, at T1 aggressive pecking was more frequent in DL than in LL birds (3.67 *vs* 1.00; P < 0.01); at T2 it tended to be more frequent in C than in DL groups (P = 0.064), and LL birds expressed this behavior significantly less than both C and DL (3.00 *vs* 0.33; P = 0.064 and 3.00 *vs* 0.33; P < 0.01, respectively). The frequency of wing flapping was lower at T1 than at both T2 and T3 (P < 0.01 and P < 0.001, respectively), but were no significant differences between T2 and

T3 (Table 4). Moreover, C groups performed this behavior more frequently than both DL and LL groups (P < 0.05 and P < 0.001). A comparable situation was observed between DL and LL birds, with DL performing wing flapping more frequently than LL (P < 0.05) (Table 4). At T1, the frequency of wing flapping was higher in DL groups than in LL (3.67 *vs* 0.00; P < 0.01), and a trend for less wing flapping was noted in LL birds compared with C (1.67 *vs* 0.00; P = 0.086). Finally, at T2 C groups exhibited significantly more frequent wing flapping than LL (2.33 *vs* 1.33; P < 001), and there was a trend for less wing flapping in DL than C (2.33 *vs* 0.33; P = 0.064) (Fig 3).

Note, outliers were considered when the data collected were not biologically possible (e.g., recording mistake), meaning that differences in animal personalities within the flock were taken into account, and this led to high standard deviations for some results. Independent of this decision, the mathematical perspective also sustained the inclusion of all outliers, due to the natural high variability among individuals [37,38].

## 3.2. Ethological tests

Neither the TI test nor analysis of the TI induction attempts revealed any significant differences between enrichment types, or a significant E × T interaction (Table 5). However, when the percentage of birds immobilized on the second attempt were compared, the values at T1 were significantly lower than at T2 (P < 0.01) and tended to be lower than at T3 (P = 0.070) (Table 5). The distribution of the data collected in the AD test showed no birds within 1 m of the operator or within 1–2 m for all the experimental groups at T0. Therefore, this sampling time was not considered in the statistical analyses to reduce the uncertainty of the model's adaptation to the dataset provided. The AD test was influenced by the enrichment type, sampling time, and their interaction (but only by the latter for the > 2 m distance). In more detail, fewer birds tended to be observed within 1 m of the operator in C groups compared with DL groups (P = 0.065). A trend was also detected in relation to sampling time, with fewer birds detected within 1 m at T1 *vs* T2 (P = 0.053) (Table 5). Considering the 1–2 m distance, fewer birds were seen at this distance from the operator at T1 than at T2 (P < 0.001). Similarly, fewer C birds were observed within the 1–2 m range than DL and LL birds (P < 0.001), while no differences were recorded between the DL and LL groups. Considering the > 2 m distance,

**Table 5. Effects on ethological tests and stress parameters of feeding a slow growing autochthonous chicken breed dehydrated and live black soldier fly larva, including enrichment treatment (E), time (T), and their interactions (E × T) (n = 6).**

| Item | Enrichment (E) | | | Time (T) | | | SEM | | P-value | | |
|---|---|---|---|---|---|---|---|---|---|---|---|
| | C | DL | LL | T0 | T1 | T2 | E | T | E | T | E × T |
| AD test < 1 m | 0.00 | 0.67 | 0.47 | – | 0.00 | 0.67 | 0.22 | 0.17 | 0.14 | 0.001 | 0.550 |
| AD test 1-2 m | 0.00[b] | 2.00[a] | 1.83[a] | – | 0.00 | 1.3 | 0.30 | 0.21 | 0.001 | <0.001 | 0.803 |
| AD test > 2 m | 7.83[a] | 4.22[b] | 4.59[b] | – | 7.08 | 4.02 | 0.43 | 0.47 | <0.001 | <0.001 | <0.001 |
| TI 1° attempt | 55.4 | 53.1 | 54.1 | 55.4 | 55.4 | 51.9 | 7.24 | 4.43 | 0.975 | 0.745 | 0.002 |
| TI 2° attempt | 54.6 | 48.7 | 46.2 | 40.2[b] | 60.1[a] | 50.9[ab] | 4.56 | 3.81 | 0.389 | 0.024 | 0.165 |
| TI 3° attempt | 33.3 | 33.3 | 38.2 | 38.2 | 33.3 | 33.3 | 1.49 | 1.49 | 0.085 | 0.085 | 0.085 |
| TI, min | 2.64 | 2.67 | 2.60 | 2.61 | 2.65 | 2.65 | 0.07 | 0.08 | 0.790 | 0.893 | 0.148 |
| ECM, ng/g | 230.8[a] | 195.5[b] | 176.5[c] | 185.8[c] | 200.2[b] | 214.1[a] | 5.21 | 5.46 | <0.001 | 0.004 | <0.001 |
| H/L ratio | 0.72 | 1.11 | 0.84 | – | – | – | 0.18 | – | 0.051 | – | – |

Abbreviations: C, control groups; DL, groups receiving supplement of dehydrated black soldier fly larvae; LL, groups receiving supplement of live black soldier fly larvae; T0, time 0; T1, time 1; T2, time 2; AD, avoidance distance test; TI; tonic immobility; ECM, excreta corticosterone metabolites; H/L, heterophile/lymphocyte ratio. Superscript letters a, b, and c indicate significant differences at P ≤ 0.05.

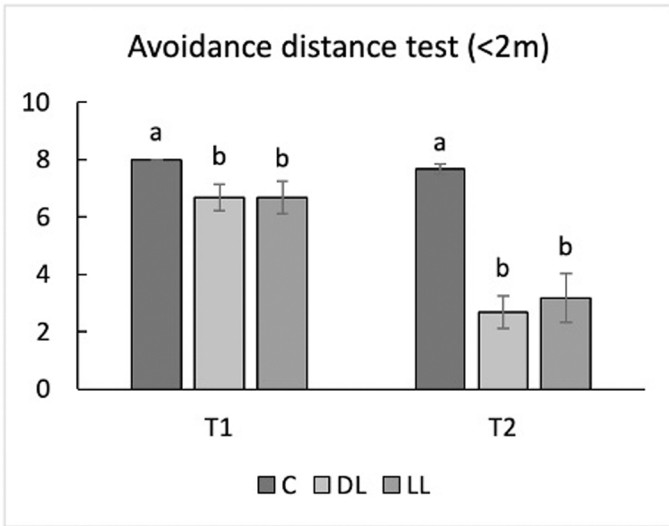 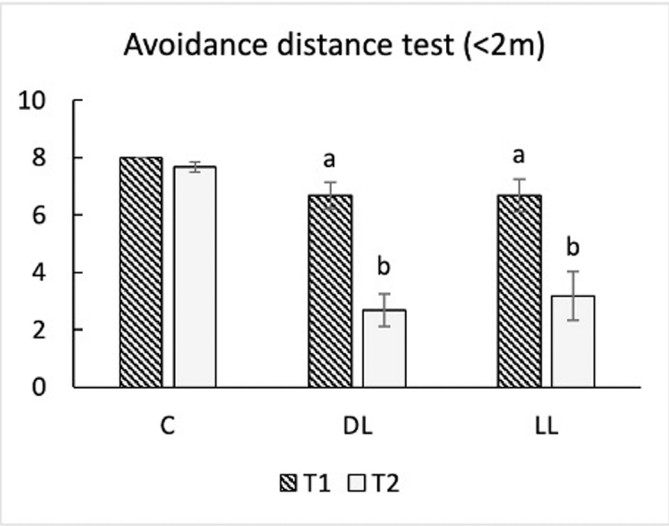

**Fig 4. The effects on the ethological tests performed on a slow growing autochthonous chicken breed fed dehydrated and live black soldier fly larva, including the interaction enrichment × time (E×T).** The differences between enrichment groups at each sampling time are displayed on the left; the differences between the sampling times within the same enrichment group are displayed on the right (n = 6). Abbreviations: T0, time 1; T1, time 2; T2, time 3; C, control groups; DL, groups supplemented with dehydrated BSFL; LL, groups supplemented with live BSFL. The superscript letters a, b, and c indicate significant differences (P < 0.05).

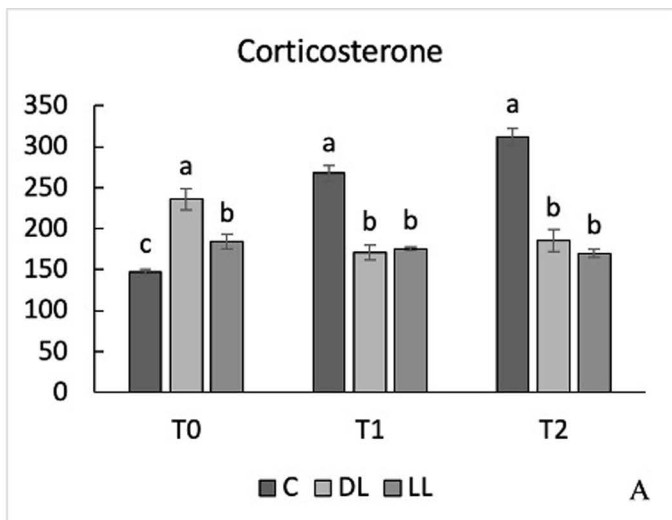 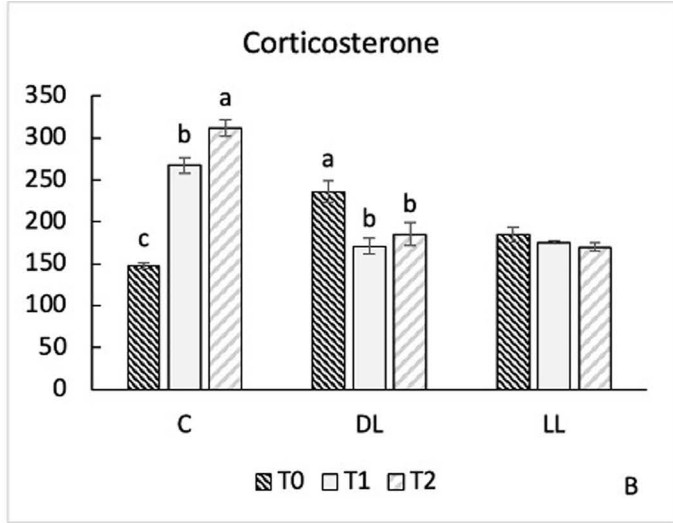

**Fig 5. The effects on corticosterone levels in a slow growing autochthonous chicken breed fed dehydrated and live black soldier fly larva, including the interaction enrichment × time (E×T).** The differences between the enrichment groups at each sampling time are displayed on the left; the differences between sampling times within the same enrichment group are displayed on the right (n = 8). Abbreviations: T0, time 1; T1, time 2; T2, time 3; C, control groups; DL, groups supplemented with dehydrated BSFL; LL, groups supplemented with live BSFL. The superscript letters a, b and c indicate significant differences (P < 0.05).

the number of birds observed was significantly higher in C groups compared with DL and LL at both T1 (8.00 *vs* 6.67 and 6.67; P < 0.01 and P < 0.05, respectively) and T2 (7.67 *vs* 2.67 and 3.17; P < 0.001) (Fig 4), whereas the number of C birds within that range decreased between T1 and T2 (P < 0.001) (Table 5). The number of C animals detected > 2 m tended to be higher at T1 than at T2 (8 *vs* 7.67; P = 0.083), whereas it was significantly higher at T1 than at T2 in both DL and LL groups (DL, 6.67 *vs* 2.67 and LL, 6.67 *vs* 3.17; P < 0.001) (Fig 4).

### 3.3. Animal based welfare assessment

Plumage condition score, pecking damage scores for skin and comb, bumble foot and footpad pododermatitis occurred too infrequently to be subjected to statistical analysis.

### 3.4. Excreta corticosterone metabolites

ECM concentration was affected by enrichment type, sampling time, and their interaction. Overall, C groups displayed a higher ECM concentration than both DL and LL (P < 0.001), and DL groups showed a higher ECM concentration than LL groups (P > 0.05) (Table 5). Comparing T2 and T3, the ECM concentration in C was higher than in DL and LL chickens (T2, 267.64 *vs* 170.92 and 175.30; T3, 311.76 *vs* 185.22 and 170.04; P < 0.001), despite no differences being observed between DL and LL groups (Fig 5).

Finally, the TI results did not correlate with the ECM concentrations obtained for any of the enrichment types (P > 0.05). The sampling time influenced the results obtained, but only within the C and DL groups. Specifically, C groups showed a significant increase in ECM concentration over time (T1-T2, 147.31 *vs* 267.64; T1-T3, 147.31 *vs* 311,76; T2-T3, 267.64 *vs* 311.76; P < 0.001, P < 0.001, and P < 0.01, respectively), whereas higher values at T1 compared with T2 and T3 were only observed in DL groups (236.20 *vs* 170.92 and 185.22; P < 0.001 and P < 0.01, respectively) (Fig 5). Furthermore, at T1, C groups displayed lower ECM concentrations compared with DL and LL groups (147.32 *vs* 236.20 and 184.44; P < 0.001), while the values in LL birds were lower than in DL birds (236.20 *vs* 184.44; P < 0.001).

### 3.5. H/L ratio

Analysis of the H/L ratio did not reveal any significant differences between groups, despite the presence of a trend for lower values in C compared with LL, and in LL compared with DL (P = 0.051) (Table 5).

## 4. Discussion

Overall, the growth performance data indicated minimal differences between the birds receiving the different enrichment treatments. However, a tendency was observed in LW and ADG in favor of the birds fed BSFL compared with the C groups. Although the difference was only small, it still indicates the potential for improvements in slow-growing bird performance, in accordance with the results obtained for fast-growing broiler chickens [9], slow-growing broiler chickens [22], and turkeys [10]. The primary aim of the present research was to understand whether the provision of dehydrated BSFL - the management of which is much simplifier than that of live larvae - offered a comparable tool to live BSFL (providing enrichment but also a potential source of competition for birds) in improving animal welfare within the context of slow-growing chicken production. Therefore, our behavioral observations addressed the moment when larvae were administered.

Overall, the frequency of indoor exploratory behavior was higher in DL than LL groups, while no differences were recorded between both DL-C birds and LL-C groups. These results are in contrast with previous reports. Ipema *et al.* [9,12,13] observed an increase in broiler chicken activity levels when the birds received either live or dehydrated BSFL supplementation, and Biasato *et al.* [14] reported an increase in foraging behaviors in broiler chickens receiving a supplement of BSFL or YM larvae. However, when comparing the different studies, the genetic features of local breeds *versus* those of broiler chickens must be taken into consideration, as the latter are generally characterized by less locomotory behavior [3]. Nevertheless, Veldkamp and van Niekerk [11] reported an increase in foraging activity in turkey

poults receiving a supplement of live BSFL at the beginning of the rearing cycle, whereas the opposite was observed at 5 weeks of age. Regarding the individual categories of exploratory behavior, the frequency of L/P exploration was lower in LL than in DL birds, and the frequency of free pen exploration behavior was lower in LL *vs* C and DL birds at T1 and in LL *vs* C groups at T3. These results might be attributable to the birds' need to express exploratory behavior being totally satisfied with LL enrichment, a theory already expressed by Veldkamp and van Niekerk [11]. A different explanation could be related to the greater volume of live larvae, the ingestion of which could have led to a sensation of satiety and thus an interruption of explorative behavior in search of food. Similarly, Tahamtani *et al.* [18] hypothesized that an incremented gastric load could be related to the chitin content of LL provision. The fact that LL birds kept exploring more after BSFL consumption suggests that the combination of insect composition and water content could produce different behavioral responses in chickens, due to the different larvae encumbrance which LL and DL may produce. Moreover, the birds of all groups displayed a reduction in the frequency of free pen exploration between T1 and both T2 and T3. Slow-growing hybrids are renowned for their reactivity, which is undoubtedly higher than that of fast-growing commercial lines [14,39]. Therefore, a reduction in overall bird exploration was not expected, as the growth of these animals is characterized by their well-proportioned body structure, which does not limit birds in their movement. However, the hot season may have affected the birds' propensity to explore [40]. Consistent with this theory, just a few birds were observed to explore the outdoor area (making it impossible to perform statistical analyses), and only the C groups exhibited a reduction in physical activity, attributable to the lack of attractive stimuli and thus awards for the birds' physical effort [41]. Moreover, we observed a decrease in the frequency of free pen exploration in DL groups at T2-T3 with respect to T1, whereas a smaller difference was observed in LL groups, but only between T1 and T3, probably due to the greater mitigation effect of heat-related sedentarism in LL birds [41]. Within the exploratory behavioral categories, the frequency of free pen exploration at T3 was higher in C than in both DL and LL, probably due to the enriched groups being occupied with larvae consumption. The L/P exploration decreased in all groups from T1 to T2, while similar results were found between T2 and T3. The introduction of a novel stimulus (plate plus larvae) into the pen may have initially stimulated the chicks' curiosity and interest [42,43]. At the same time, once the birds had become used to the object's presence and its associated benefits (larvae ingestion) identified by the LL and DL birds, they may have become less attracted to the object itself (i.e., the plate) following larvae consumption, explaining the reduction in L/P exploration at the T2 and T3. Overall, the frequency of agonistic behavior frequency was inconsistent over time, with the lowest levels observed at T2, and no differences recorded between T1 and T3. We categorized the agonistic behaviors for the comparisons between different age groups. It must be considered that social interactions in juvenile chicks are largely affected by play dynamics, with no apparent immediate function or benefits to be gained; they are considered a general preparation for adulthood, and thus fundamental for the proper development of the birds' psychological and physical skills [44,45]. Behaviors such as sparring (play fighting) are mostly involved in play dynamics in juvenile chickens [44,46,47]. In concomitance with the chickens' development, some components of play may change into agonistic behaviors, with sparring mutating into fighting over time, as shown by Baxter *et al.* [46], although the incidence of these behaviors in chicks does not predict that of aggressive behaviors in adulthood [4]. In the BS breed, aggressive interactions are observed at early life stages. However, further research is needed to better understand when play behaviors transition into aggressive interactions. The decrease in the frequency of aggressive behaviors at T2 (7 weeks of age) occurred during the period of hierarchy establishment (6–12 weeks) [4,48], event that could have had a positive effect on the birds' interactions. The subsequent increase

in agonistic behavior displayed at T3 could instead be related to sexual maturity [5], which occurs in BS males around 20–24 weeks of age [7].

Delving into the statistical interaction effects, the DL groups expressed more agonistic behavior at T1 than at T2, while it was less frequent at T2 than at T3. As observed for explorative behaviors, agonistic patterns were well mitigated by BSFL administration, with a reduction in aggressive behavior seen in LL groups over time (considered the biological value 1.00 *vs* 1.33), and a lower value at T1 compared with DL groups (a trend only due to the high standard deviation present), probably due to the high motility of the live larvae [9,17,43]. This behavioral response was reflected in both categories of agonistic behavior: aggressive pecking (LL<DL groups at T1; LL<C, DL<C groups -trend- at T2) and wing flapping (LL<DL groups at T1, LL<C -trend- at T1; LL<C, DL<C groups -trend- at T2), with a higher frequency of aggressive pecking in DL groups at T1 *vs* T2, possible due to more time spent consuming the larvae at T2. Aggressive pecking can be categorized according to the body area involved, being primarily associated with the head or neck of the victim [49]. In our research, the frequency of aggressive pecking was significantly lower in LL than C groups, whilst the frequency in DL birds was not significantly different to that of either LL or C. Overall, the frequency of wing flapping was lower at T1 than at either T2 or T3. Chickens may engage in wing flapping for a range of different reasons, including expressing comfort [49], territorial displays [50], aggressiveness [23], and courtship behavior [51], and it plays an important role in enhancing wing-bone strength [52]. In this study, the frequency of wing flapping was lower at T1 than at T2 or T3 in all groups, likely related to their nearing adulthood and the increases in competition between birds at that age [4,48]. In light of the present findings, further research is required to characterize the behavioral patterns of slow-growing chicken breeds better and thus the potential benefits of providing these birds with dehydrated or live BSFL as environmental enrichment.

The AD test provides an indication of a bird's level of fear towards humans, and it is considered a valid measurement of welfare [28,53]. Interestingly, at T0 (i.e., before having received any larvae) no birds ventured within 2 m of the operator. Bearing in mind the nature of chickens as a pray species, their reluctance to get within 1 m could be considered an intrinsic survival mechanism against environmental threats and predators, although a healthy equilibrium between fear and exploration are also fundamental for successful survival [2,49,54]. However, the fact that no birds ventured within 2 m, despite their daily exposure to humans, could represent a consolidated habit. This result might be traced back to the strain's feral traits, being a breed that has long been exempt from genetic selection [42,55]. In support of this hypothesis, comparable results were also reported at T0 by Bongiorno *et al.* [15] for slow-growing broilers at the 1 m and 1–2 m distances. At T1, the number of birds within the 1 m distance tended to be higher in DL than in C groups, which did not receive any larvae at all, and a further increase was also observed at T2, suggesting that as the birds became more familiar with larva supplementation, their confidence in the operator also increased. However, the average value obtained was less that one, generating data with little biological meaning. Curiously, fewer birds were counted between 1–2 m at T1 (0 specimens) than at T2 in all the experimental groups, showing the high amount of time required to develop trust towards the human operator (T1 = 91 days of age; T2 = 141 days of age) to. In the experiment carried out by Bongiorno *et al.* [15], involving a similar number of reared birds (slow-growing broilers), performed in the same facility, not only were birds already observed within 1–2 m at T0 (27 days of age), but there was no difference in the number of birds (around five) recorded at T1 (41 days of age) *vs* T3 (62 days of age). At that stage, the birds were younger than the ones in this study, but already venturing closer to the operator than our BS males, once again demonstrating the considerable behavioral differences between local breeds and genetic strains.

Furthermore, in our study, the AD test reported fewer C birds within 1–2 m of the operator than for the DL and LL groups. Moreover, the number of birds counted in the LL and DL groups at a distance of > 2 m from the operator was twice that observed in the C groups, while no differences were recorded between DL and LL groups for both 1–2 m and > 2 m. The present findings demonstrated the equal potential of dehydrated and live BSFL supplementation to reduce the birds' fear of humans and improve the human-chicken bond. Hence, BSFL can help local chicken breeds overcome their characteristic weariness and defensiveness behavior, which despite their domestication have maintained more of their ancestors' traits, characterized by a higher fear response than strains selected for their high meat or egg production [56]. Finally, the number of C birds at a distance > 2 m showed a tendency to be lower at T2 *vs* T1, whereas a significant reduction was observed in both DL and LL chickens, underscoring the positive influence of larva provision on the human-chicken relationship.

We also evaluated fearfulness in the three groups using the TI test, a renowned indicator of fear in birds, able to provide reliable results if well standardized [57–59]. However, the results obtained did not agree with those of the AD test, since no significant differences in TI were recorded between times and groups; therefore, larva administration was not found to modulate fear levels as assessed by the TI test. However, the TI test resembles a predator attack and evaluates the animals' reactions towards the event [59–61], and thus concerns a different type of fear than evaluated in the AD test. Our results also disagree with those reported by Ipema *et al.* [12] on broiler chickens fed a live BSFL supplement. The difference probably reflects the difference in the type of stimulus the birds were subjected to, namely a potential predator attack (TI test) *vs* an unobtrusive external approach (AD test). This takes on a positive meaning when applied to the farm context: the enriched birds showed more confidence towards their human operator, whilst maintaining a basal level of fear crucial for survival. Nonetheless, we cannot exclude some effects of habituation, since TI was successfully induced on the second attempt in a smaller percentage of birds at T1 than at T2 and T3 regardless of larvae administration. Indeed, this study only performed the tests three times over the course of the experiment in order to prevent animal habituation to the procedure, however, the mechanism could be affected by strain, resulting in a greater predisposition of some genotypes to habituation than others [62].

Moving to the animal-based welfare parameters, the absence of any damage to the feathers, of any skin or comb wounds, and any imparities in leg health do not concur with our theory that BSFL supplementation enhances bird welfare. Starting from the assumption that hierarchies are not usually established prior to an age of 6–10 weeks of age [48] and that the sexual maturity in the BS breed is reached at around 20–24 weeks of age [7], we had expected to find indirect evidence of aggressive behavior on the birds' bodies. Moreover, we adopted a moderate rearing density, but the presence of male birds only is an important factor, especially when it comes to native breeds characterized by an alert temperament. The good welfare conditions of the birds might be more related to the season than the space provided. Environmental temperature and relative humidity were recorded in the barn at bird height, and the temperature reached a peak of 37.5°C. This may have attenuated bird responsiveness to competition and agonistic stimuli, thus resulting in no detectable physical damage on the birds [41]. Although no correlation was found between the results for TI and ECM, interesting results were obtained in relation to the latter. The ECM provides a reliable indicator of stress in birds [63], despite stress encompass both detrimental conditions for animals' fitness and adaptative mechanism fundamental to manage and overcome daily challenges [64]. Notably, an increase in ECM levels were noted over time, but only in the C groups, which also generally displayed higher ECM values than LL and DL birds. The ECM increases may have been driven by hierarchy establishment in the flocks and/or by the increases in ambient

temperatures. In support of this hypothesis, previous research has demonstrated the capacity of environment enrichment to modulate the effects of stressors in poultry [65]. Curiously, the highest ECM values were obtained in DL groups, whereas the lowest corticosterone concentration was obtained in C groups at T1. Since the management practices did not vary between birds, any differences detected prior to the start of larva provision must be related to individual variations and random factors. However, this basal level enhanced the capacity of larva administration to attenuate stress levels; indeed, after the start of larva administration the highest ECM levels were obtained in C birds, whereas the lowest were detected in DL then LL groups. In contrast with these findings, an opposite trend was recorded for the H/L ratio, with higher values recorded in DL *vs* LL *vs* C. The H/L ratio is considered a reliable measurement of stress levels in chickens, providing long-term information on the birds' condition [66,67]. However, the H/L ratio may also vary among genotypes [68] and can be affected by restraint procedures [68,69], even though all the birds in the trial received the same management treatments. There are no data in the literature regarding the possible effects of BSFL administration on H/L concentrations, something that will need to be addressed by future research. Recent studies reported how the physico-chemical characteristics and microbial content of BSFL can vary in relation to pre-treatments and dehydration procedures [70]. As the H/L ratio also relates to the level of immune system activity [71], an effect of the BSFL microbial load on this parameter cannot be excluded. Another consideration emerged when we compared the present results to a previous study conducted by our team [15], where similar results for the H/L ratio were found in slow-growing broilers receiving a 10% supplement of live BSFL, which were significantly higher compared with those in C birds. One possible explanation, mentioned in the above-cited paper, could be the chickens' anticipation for larva administration, with greater levels of excitement exhibited for BSFL than C groups.

## 5. Conclusions

Live and dehydrated BSFL supplementation resulted in comparable effects on several parameters investigated in this experiment, providing promising insights into their potential benefits. It is important to consider that enrichment capable of stimulating physical activity, such as live larvae, may not be as impactful in species where such behavior is already well-developed, such as in fast-growing broiler chickens. However, in the case of slower-growing, local breeds like the BS breed, both live and dehydrated BSFL showed similar positive effects, suggesting that both forms can be effective. While the study was conducted with a limited number of birds, due to the conservation efforts for the BS breed, the findings still provide valuable starting points for future research into dual-purpose genotypes with similar behavioral characteristics. Increasing the sample size in future studies would enhance the generalizability of the results and further solidify these findings.

Overall, this study contributes valuable information to the understanding of how BSFL supplementation can influence exploratory behavior in chickens. The motility of live larvae likely provided greater satisfaction for the birds, although additional research is needed to explore the potential gastrointestinal effects of live larvae on exploratory activity. Furthermore, supplementing birds with whole BSFL—whether live or dehydrated—may strengthen the human-animal bond, which is particularly beneficial in local breeds that are generally more cautious of humans than highly selected strains.

The similar effects of live and dehydrated BSFL supplementation on stress modulation, particularly during heat and competition, highlight the potential of BSFL to enhance animal welfare and performance, especially in slow-growing chicken genotypes. This study lays the

groundwork for future investigations into the role of BSFL in improving the management and well-being of chickens, with exciting possibilities for both commercial and conservation-focused poultry breeding programs.

Live and dehydrated BSFL supplementation resulted in comparable effects on several parameters investigated in this experiment. However, this finding must be considered according to the context. It is comprehensible that enrichment able to stimulate physical activity in birds, ending in a reward being given (live larvae), may not have a positive effect in animals where this capability is not compromised during growth, as occurs in fast-growing broiler chickens. In this case, it is reasonable that equivalent effects were observed in the DL and LL groups. Moreover, increasing the number of birds involved in the experiment could help produce a more representative evaluation of the behaviors considered. The BS breed nearly became extinct just a few years ago (but was saved thanks to a conservation program carried out at the University of Turin), and the number of specimens recruited for the study was limited by the number of birds available in the region. However, our findings should be considered a starting point for other studies into dual-purpose genotypes characterized by similar behavioral traits. Interesting conclusions can still be drawn from this study. Both live and dehydrated BSFL supplementation were found to influence explorative activity in these chickens, with greater satisfaction probably associated with live rather than dehydrated larvae by virtue of their motility, although further research is required to investigate the gastro-intestinal encumbrance role of LL on exploration levels in birds. In addition, feeding birds whole BSFL could advantageously reinforce the human-animal bond, regardless of the larvae's form (live or dehydrated). This may have beneficial effects on bird daily management practices, especially in local breeds which are generally more fearful of humans than highly selected strains. Moreover, the effects of live and dehydrated BSFL supplementation on ECM were similar. The data presented here also offer new insights into the modulating effects of BSFL supplementation on stress induced by heat and bird competition during the growth-phase of slow-growing chicken genotypes.

## Supporting information

**S1 File. Bongiorno et al. 2024.**
(DOCX)

## Acknowledgments

The authors acknowledge the support provided by INAGRO (Belgium) and ENTOMO (Spain) technicians for the live and dehydrated BSFL provision, and are thankful to Mr. Dario Sola for his support in chicken management.

## Author contributions

**Conceptualization:** Edoardo Fiorilla, Marta Gariglio, Achille Schiavone.

**Formal analysis:** Valentina Bongiorno.

**Funding acquisition:** Francesco Gai, Achille Schiavone.

**Investigation:** Valentina Bongiorno, Edoardo Fiorilla, Marta Gariglio, Valeria Zambotto, Eleonora Erika Cappone, Stefania Bergagna, Isabella Manenti, Elisabetta Macchi.

**Methodology:** Valentina Bongiorno, Edoardo Fiorilla, Marta Gariglio, Francesco Gai, Achille Schiavone.

**Supervision:** Francesco Gai, Achille Schiavone.

**Validation:** Achille Schiavone.

**Visualization:** Achille Schiavone.

**Writing – original draft:** Valentina Bongiorno.

**Writing – review & editing:** Valentina Bongiorno, Edoardo Fiorilla, Marta Gariglio, Francesco Gai, Achille Schiavone.

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
