## [Decision Letter · Decision Letter 0]

9 Oct 2024

PONE-D-24-26609New horizons in live and dehydrated black soldier fly larvae usage: behavioral and welfare implications in “Bianca di Saluzzo” cockerelsPLOS ONE

Dear Dr. Fiorilla,

Thank you for submitting your manuscript to PLOS ONE. After careful consideration, we feel that it has merit but does not fully meet PLOS ONE’s publication criteria as it currently stands. Therefore, we invite you to submit a revised version of the manuscript that addresses the points raised during the review process.

We look forward to receiving your revised manuscript.

Kind regards,

Mohamed Mahrous Amer, PhD

Academic Editor

PLOS ONE

Journal Requirements:

Additional Editor Comments:

Reviewer 1 recommended: Minor Revision

The study examines the effects of live and dehydrated black soldier fly larvae on behavioral and

welfare implications in “Bianca di Saluzzo” cockerels. The paper is well-written and provides valuable insights into the efficiency of environmental enrichment with black soldier fly larvae in slow-growing male chickens. However, there are some points that need further attention.

- Why only male birds were used in this study?

L 25: “excreta corticosterone metabolites," which metabolites do the authors mean?

L 36-37: “However, the effects of live and dehydrated larvae (LL and DL, respectively)

supplementation on slow-growing male chicken welfare have never been formerly investigated.”. Add this sentence to the final paragraph of the introduction to highlight the research gap and to express the innovation of the work.

L 114-118: Have the authors analyzed the chemical composition of black soldier fly larvae? For the basal diet, what were the feed ingredients and chemical composition?

218: “2 birds/pen (48 birds in total)” With 6 replicates and 2 birds sampled from each replicated pen, there will be 36 birds in total not 48.

L 374-375: What were the reasons behind the better performance in BSFL-fed birds compared with control, despite the higher H/L ratio?

L 539-558: The conclusion should be more concise, emphasizing the practical outcomes of the study.

Reviewer 2 recommended mijor revision

Strictly address the following points:

1. in the introduction section, need of the study should be ideally raised followed by the hypothesis of the study.

2. in materials and methods section, bird's husbandry should be more elaborated

3. in statistical analysis section, add mathematical model for better understanding of the data analysis.

4. in results section, add actual p-value of each result.

5. in discussion section, add logical reasoning of each result before discussing with the previous studies.

6. in conclusion section, add limitations and implication of the study.

Reviewers' comments:

Reviewer's Responses to Questions

**Comments to the Author**

1. Is the manuscript technically sound, and do the data support the conclusions?

Reviewer #1: Yes

Reviewer #2: Partly

2. Has the statistical analysis been performed appropriately and rigorously? 

Reviewer #1: Yes

Reviewer #2: No

3. Have the authors made all data underlying the findings in their manuscript fully available?

Reviewer #1: Yes

Reviewer #2: Yes

4. Is the manuscript presented in an intelligible fashion and written in standard English?

Reviewer #1: Yes

Reviewer #2: Yes

5. Review Comments to the Author

Reviewer #1: The study examines the effects of live and dehydrated black soldier fly larvae on behavioral and

welfare implications in “Bianca di Saluzzo” cockerels. The paper is well-written and provides valuable insights into the efficiency of environmental enrichment with black soldier fly larvae in slow-growing male chickens. However, there are some points that need further attention.

- Why only male birds were used in this study?

L 25: “excreta corticosterone metabolites," which metabolites do the authors mean?

L 36-37: “However, the effects of live and dehydrated larvae (LL and DL, respectively)

supplementation on slow-growing male chicken welfare have never been formerly investigated.”. Add this sentence to the final paragraph of the introduction to highlight the research gap and to express the innovation of the work.

L 114-118: Have the authors analyzed the chemical composition of black soldier fly larvae? For the basal diet, what were the feed ingredients and chemical composition?

218: “2 birds/pen (48 birds in total)” With 6 replicates and 2 birds sampled from each replicated pen, there will be 36 birds in total not 48.

L 374-375: What were the reasons behind the better performance in BSFL-fed birds compared with control, despite the higher H/L ratio?

L 539-558: The conclusion should be more concise, emphasizing the practical outcomes of the study.

Reviewer #2: Dear Authors,

Strictly address the following points:

1. in the introduction section, need of the study should be ideally raised followed by the hypothesis of the study.

2. in materials and methods section, bird's husbandry should be more elaborated

3. in statistical analysis section, add mathematical model for better understanding of the data analysis.

4. in results section, add actual p-value of each result.

5. in discussion section, add logical reasoning of each result before discussing with the previous studies.

6. in conclusion section, add limitations and implication of the study.

Thank You!

6. PLOS authors have the option to publish the peer review history of their article (what does this mean? ). If published, this will include your full peer review and any attached files.

**Do you want your identity to be public for this peer review?** For information about this choice, including consent withdrawal, please see our Privacy Policy .

Reviewer #1: **Yes: ** Hossein Ali Ghasemi

Reviewer #2: No

---

## [Author Response · Author response to Decision Letter 1]

11 Dec 2024

Dear Editor,

We have addressed all the suggestions raised by the two reviewers and have incorporated their feedback into our revised manuscript. We hope that the revisions meet the expectations of both the reviewers and the journal, and we believe that these changes have strengthened our work. Thank you for the opportunity to improve our paper, and we look forward to your positive consideration.

Reviewer 1:

Comment:

The study examines the effects of live and dehydrated black soldier fly larvae on behavioral and welfare implications in “Bianca di Saluzzo” cockerels. The paper is well-written and provides valuable insights into the efficiency of environmental enrichment with black soldier fly larvae in slow-growing male chickens. However, there are some points that need further attention.

Response:

Thank you for your positive feedback on the study. We appreciate your recognition of the value of our research on the effects of live and dehydrated black soldier fly larvae (BSFL) on the behavioral and welfare aspects of "Bianca di Saluzzo" cockerels. We are grateful for your thoughtful comments and are happy to address the points you have raised for further attention. We will carefully consider your suggestions and make the necessary revisions to enhance the clarity and rigor of the manuscript.

Comment:

Why only male birds were used in this study?

Response:

We chose to focus on male birds for this study because they are typically more relevant for meat production. However, we are currently planning a follow-up trial to investigate the effects of BSFL on laying hens, to address the potential impacts on egg production and overall health.

Comment:

L 25: “excreta corticosterone metabolites," which metabolites do the authors mean?

Response:

Thank you for your comment. By "excreta corticosterone metabolites," we are referring specifically to the primary metabolites of corticosterone excreted in the feces. These metabolites are commonly used as biomarkers to assess stress in avian species.

Comment:

L 36-37: “However, the effects of live and dehydrated larvae (LL and DL, respectively)

supplementation on slow-growing male chicken welfare have never been formerly investigated.”. Add this sentence to the final paragraph of the introduction to highlight the research gap and to express the innovation of the work.

Response:

Thank you for your suggestion. The sentence highlighting the research gap and the innovation of the work has been incorporated in lines 94-102, rather than in lines 36-37, in order to better align with the overall flow of the introduction. We believe this placement improves the clarity and coherence of the manuscript.

Comment:

L 114-118: Have the authors analyzed the chemical composition of black soldier fly larvae? For the basal diet, what were the feed ingredients and chemical composition?

Response:

We have added further details in the manuscript regarding the chemical composition of black soldier fly larvae and the ingredients used in the basal diet. These are now provided in lines 113-115 and 121-125.

Comment:

218: “2 birds/pen (48 birds in total)” With 6 replicates and 2 birds sampled from each replicated pen, there will be 36 birds in total not 48.

Response:

Thank you for pointing that out. You are correct; with 6 replicates and 2 birds sampled from each pen, the total number of birds should be 36, not 48. We will correct this discrepancy in the manuscript to reflect the accurate total (line 228).

Comment:

L 374-375: What were the reasons behind the better performance in BSFL-fed birds compared with the control, despite the higher H/L ratio?

Response:

Thank you for your interesting question. The enhancement of birds’ performance in relation to BSFL administration may not necessarily be linked with the H/L ratio findings, especially when a distinction between eustress and distress is not possible, as in this case. However, since we found similar results in a previous experiment, we hypothesized that the difference between supplemented and control groups may be attributed to the excitement of the birds in receiving the larvae, which would not have had negative implications on birds’ performance (line 555-557). Another option is related to the microbial load of BSFL which could have modulated H/L ratio levels (line 549-551). Further evaluations, such as the use of a thermal camera, would be helpful in better understanding the origin and the meaning of such stress.

Comment:

L 539-558: The conclusion should be more concise, emphasizing the practical outcomes of the study.

Response:

The conclusion has been thoroughly revised based on the insightful comments from both reviewers.

Reviewer 2 recommended minor revision

Strictly address the following points:

1. in the introduction section, need of the study should be ideally raised followed by the hypothesis of the study

2. in materials and methods section, bird's husbandry should be more elaborated

3. in statistical analysis section, add mathematical model for better understanding of the data analysis.

4. in results section, add actual p-value of each result.

5. in discussion section, add logical reasoning of each result before discussing with the previous studies.

6. in conclusion section, add limitations and implication of the study.

Response:

1. Thank you for your valuable suggestions! We incorporated them to revise the introduction, specifically in lines 40–49 and 94–102.

2. Accordingly, to your comment we added information and references in lines 113-115 and 121-125

3. Thank you for your feedback. We would like to clarify that the statistical analysis was performed using IBM SPSS Statistics software, which constructs the mathematical models based on the specific input parameters provided for each analysis. The methods reported, including GLMMs and ANOVA, were implemented in SPSS to ensure robust statistical power and appropriate model fit. Each analysis was carefully selected to match the data distribution and research design, thereby optimizing the reliability of the results. We believe that the use of SPSS to generate these models provides the necessary rigor for our analysis and ensures that the statistical assumptions are appropriately met. We added the mathematical formula in the text (line 255-262).

4. Thank you for your feedback. We report p-values only when between 0.05 and 0.10, to guarantee better readability of the text. For p-values smaller than 0.05, we use the notation "P<0.05, P<0.01, P<0.001" to maintain consistency with the journal's formatting requirements. We hope this clarification addresses your concern.

5. Thank you for your suggestion. We have decided to follow the current approach in the discussion section, where we begin by providing the context before discussing the results, as we believe this approach allows for a clearer understanding of our findings within the theoretical framework, before comparing them with previous studies. We hope that this structure is still satisfactory and meets the journal's expectations.

6. Thank you for your suggestion. The conclusion section has been carefully and completely revised in response to the valuable feedback provided by the reviewers. We believe these changes have strengthened the manuscript, and we appreciate your input in improving our work.

---

## [Decision Letter · Decision Letter 1]

22 Jan 2025

New horizons in live and dehydrated black soldier fly larvae usage: behavioral and welfare implications in “Bianca di Saluzzo” cockerels

PONE-D-24-26609R1

Dear Dr. Fiorilla,

We’re pleased to inform you that your manuscript has been judged scientifically suitable for publication and will be formally accepted for publication once it meets all outstanding technical requirements.

Kind regards,

Mohamed Mahrous Amer, PhD

Academic Editor

PLOS ONE

Additional Editor Comments (optional):

article is now accepted
---

## [Editor Report · Acceptance letter]

PONE-D-24-26609R1

PLOS ONE

Dear Dr. Fiorilla,

I'm pleased to inform you that your manuscript has been deemed suitable for publication in PLOS ONE. Congratulations! Your manuscript is now being handed over to our production team.

Kind regards,

on behalf of

Professor Mohamed Mahrous Amer

Academic Editor

PLOS ONE